

# The disappearing hand: vestibular stimulation does not improve hand localisation

Luzia Grabherr[1,2,*], Leslie N. Russek[1,3,*], Valeria Bellan[1,4], Mohammad Shohag[1], Danny Camfferman[1] and G. Lorimer Moseley[1]

[1] School of Health Sciences, University of South Australia, Adelaide, SA, Australia
[2] Psychiatric Liaison Service, University Hospital of Lausanne, Lausanne, Switzerland
[3] Clarkson University, Physical Therapy Department, Potsdam, NY, USA
[4] Department of Psychology, University of Milano-Bicocca, Milan, Italy
* These authors contributed equally to this work.

## ABSTRACT

**Background:** Bodily self-consciousness depends on the coherent integration of sensory information. In addition to visual and somatosensory information processing, vestibular contributions have been proposed and investigated. Vestibular information seems especially important for self-location, but remains difficult to study.

**Methods:** This randomised controlled experiment used the MIRAGE multisensory illusion box to induce a conflict between the visually- and proprioceptively-encoded position of one hand. Over time, the perceived location of the hand slowly shifts, due to the fact that proprioceptive input is progressively weighted more heavily than the visual input. We hypothesised that left cold caloric vestibular stimulation (CVS) augments this shift in hand localisation.

**Results:** The results from 24 healthy participants do not support our hypothesis: CVS had no effect on the estimations with which the perceived position of the hand shifted from the visually- to the proprioceptively-encoded position. Participants were more likely to report that their hand was 'no longer there' after CVS. Taken together, neither the physical nor the subjective data provide evidence for vestibular enhanced self-location.

Corresponding author
Luzia Grabherr,
luziagrabherr@gmx.ch

## INTRODUCTION

The important contribution of accurate and congruent sensory information to bodily self-consciousness has been widely acknowledged (*Blanke, 2012*; *Blanke & Metzinger, 2009*; *Serino et al., 2013*). Besides visual and somatosensory information, vestibular contributions have also been suggested and investigated. Particularly, research in patients involving altered bodily awareness (e.g. neglect, somatoparaphrenia, phantom limb sensations) suggests that vestibular stimulation can attenuate dysfunctional bodily self-consciousness (for reviews see *Grabherr, Macauda & Lenggenhager, 2015*; *Lopez, 2015*;

*Mast et al., 2014*). Patients with impaired vestibular function on the other hand are prone to distorted own body representations (*Lopez et al., 2018*) and out-of-body experiences (*Lopez & Elzière, 2018*).

Bodily self-consciousness is thought to consist of at least three different components: body ownership (the experience of owning a body), first-person perspective (the experience of taking a visuospatial perspective from that body), and self-location (the experience of being a body in a given location) (*Blanke & Metzinger, 2009*; *Serino et al., 2013*). Recently, *Serino et al. (2013)* have proposed that bodily ownership and self-location rely on partially distinct neural correlates, that is, the premotor cortex and the temporo-parietal junction, respectively. The temporo-parietal junction is likely a key player in the integration of vestibular and other sensory cues (*Ionta et al., 2011*; *Pfeiffer et al., 2016*; see also *Gallace & Bellan, 2018*). Given that the vestibular system provides important information about body position and self-motion in space, it has been proposed that vestibular information is particularly relevant to self-location (*Lenggenhager & Lopez, 2016*; *Pfeiffer et al., 2013*; *Serino et al., 2013*).

Bodily illusions provide clever experimental tools because they can disrupt different components of bodily self-consciousness also in healthy participants. For example, during the rubber hand illusion (RHI) (*Botvinick & Cohen, 1998*), participants are asked to rest their hands on a table. One of their hands is hidden and 'replaced' by a prosthetic (rubber) hand. The experimenter then starts stroking the rubber hand and the participants' hidden hand synchronously. After a few seconds, participants start to perceive that the rubber hand belongs to them. In addition, when asked to localise their hidden hand, they point at a spot between the real and the rubber hand. This localisation error is called 'proprioceptive drift' and it has been correlated with the strength of the illusion (*Botvinick & Cohen, 1998*; *Tsakiris & Haggard, 2005*; but also see, for example, *Rohde, Di Luca & Ernst, 2011*). The MIRAGE multisensory illusion box allows the manipulation of self-location of a body part specifically by using a compelling perceptual illusion called the disappearing hand trick (DHT) (*Newport & Gilpin, 2011*). Participants see a video in real-time of their hands moving inwards, while, in fact their hands are moving outwards. Therefore, when the hands are placed on the bottom of the MIRAGE box, in line with the visual input, participants believe that their hands are close together, while they are in fact far apart. This is due to an induced sensory mismatch of up to 14 cm, between the proprioceptively-encoded location of the participants' hidden hand (where the participants feel the hand to be), and its visually-encoded location (where the participants saw their hand for the last time), while participants remain unaware of this mismatch (*Bellan et al., 2017a*, *2015*) (see Fig. 1). Recent research shows that self-location of the hand drifts over time from the last seen location towards the actual physical location, without the participants realising. This sensory mismatch allows testing of the relative weighting of visual and proprioceptive information in generating the perceived location of the hand, while minimising confounding effects introduced by expectation or report bias (*Bellan et al., 2017a*, *2015*).

Thus far, studies in healthy participants have suggested that caloric vestibular stimulation (CVS, please refer to the method section 'CVS' for more information on this
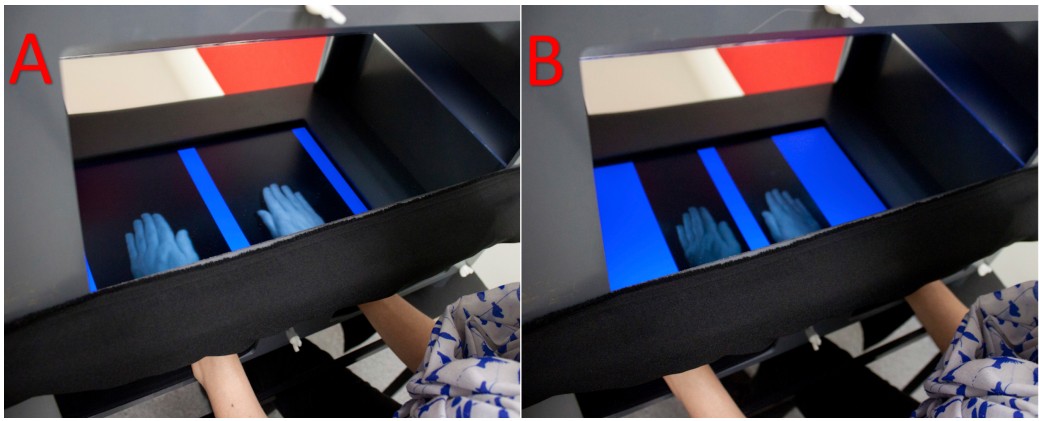

**Figure 1 Disappearing hand procedure using the MIRAGE multisensory illusion box.** Depicted is the participant's perspective while watching her hands during the adaptation procedure inside the MIRAGE box at the beginning of the procedure (A) and towards the end (B). The actual hand end position is further outwards compared to the seen position. Please note that these pictures are only for illustrating purposes; the participants did not actually see their elbows, which were occluded from view with a bib to prevent visual cues about the actual hand position. As indicated during the debriefing, most participants were not aware of the incongruence between the visual and physical hand position. Photo credit: Cat Jones.                    

technique) and in particular left cold CVS (predominantly activating the right hemisphere) can enhance somatosensory perception (i.e. tactile perception) in the left and right hand (*Ferrè, Bottini & Haggard, 2011*; *Ferrè et al., 2013*). Another study using left anodal and right cathodal galvanic vestibular stimulation (GVS, which also predominantly activates the right hemisphere) reported reduced mislocalisation of one's own hand (i.e. enhanced proprioception) in the RHI (*Ferrè, Berlot & Haggard, 2015*) leading to the proposition that 'vestibular signals rebalance sensory events away from the world and towards the self' (*Ferrè & Haggard, 2016*: p. 77). This implies that vestibular stimulation may enhance weighting of proprioceptive information during the sensory mismatch induced via MIRAGE.

However, given the strong interrelation of visual and vestibular signals, some factors should be accounted for. First, it is also plausible that vestibular stimulation would enhance the weighting of visual information. Relevant to this possibility are results from other RHI experiments, suggesting that vestibular stimulation enhances visual capture and not proprioceptive information. *Ponzo et al. (2018)* found an enhanced proprioceptive drift towards the rubber hand. Although *Lopez, Lenggenhager & Blanke (2010)* found no effect of vestibular stimulation on the proprioceptive drift, they did find increased illusory ownership of the rubber hand and location of touch evidenced by questionnaire responses. In a subsequent experiment using a non-visual variant of the RHI, vestibular stimulation affected neither subjective feelings of body ownership nor proprioceptive drift, suggesting no influence of vestibular stimulation when visual feedback is absent (*Lopez, Blanke & Mast, 2012*). If this were the case (i.e. that vestibular stimulation affects only on visual capture), we would observe a slower drift and/or a starting location that is closer to the visual hand location, when compared to a sham condition. Second,

vestibular stimulation-induced attention effects have been debated. Potential vestibular attentional effects are typically observed towards the ipsilateral side of the stimulated ear. Thus, left cold CVS (as used in our study) could potentially lead to a generic attentional shift towards the left hemispace. However, such effects are critically debated and even when they are observed, they are usually short-lived in healthy participants; therefore, we did not predict such effects (see *Bottini et al., 2013*, for a critical discussion). In order to rule out such effects, we tested both hands, because they would be affected differently (the initial location and/or drift of the right hand would be shifted towards the proprioceptively-encoded hand position, whereas the initial location and/or drift of the left hand would be shifted towards the visually-encoded hand position).

Clarifying the potential role of the vestibular system in bodily self-consciousness is important because vestibular stimulation has been proposed for treating a wide range of conditions involving body awareness, despite a lack of strong supportive evidence (for critical reviews see *Grabherr, Macauda & Lenggenhager, 2015*; *Miller, 2016*). Findings from this study may help further the understanding of the role of vestibular information in self-location, particularly regarding the prioritisation of conflicting sensory input (i.e. visual vs. proprioceptive). This knowledge may help explain underlying mechanisms responsible for observed vestibular induced effects in patients with impaired bodily self-consciousness.

Taken together and despite some contradictory findings, we hypothesised that left cold CVS augments the self-location component of bodily self-consciousness. This should be reflected by a more pronounced shift from visual to proprioceptive location in the MIRAGE illusion, such that vestibular stimulation compared to a sham condition leads to (i) a faster drift in perceived location, towards the proprioceptively-encoded hand position, (ii) an initial perceived location that is closer to the proprioceptively-encoded hand position, or (iii) both.

## MATERIALS AND METHODS
### Participants
An a priori power analysis was performed to determine the number of participants. Given the use of a novel tool to assess hand localisation after CVS, it was difficult to calculate effect sizes based on previous data. Therefore, we assumed a generic medium to large effect size of $f = 0.3$ with a power $= 0.8$. Estimations were performed with MorePower 6.0 (*Campbell & Thompson, 2012*) detecting a required sample of 22 participants. We recruited 24 to allow for a $2 \times 2$ counterbalanced design. This sample size is slightly bigger than previous studies testing the influence of vestibular stimulation on bodily self-consciousness and body representation but using different tasks (*Lopez, Lenggenhager & Blanke, 2010*; *Lopez et al., 2012*). Participants were recruited through noticeboards and social media. The mean age was 25 ± 9 SD; 14 were female. One participant was replaced because no CVS response could be evoked (no nystagmus and no reported subjective effects). All participants needed to be right-handed according to a shortened version of the Edinburgh Handedness Inventory (*Veale, 2014*) and had normal or corrected-to-normal vision. Participants were assessed with a short medical history questionnaire and were
excluded if they reported otological, neurological, or psychological problems. A physician verified the integrity of the eardrum (otoscopy) before and after the experiment. Participants received financial compensation for their participation.

The study was approved by the local ethics committee (University of South Australia, Application ID: 32955) and was performed in accordance with the Declaration of Helsinki. Written informed consent was obtained before commencing the experiment.

## Caloric vestibular stimulation

Caloric vestibular stimulation was performed with participants lying supine with their head positioned 30° upright, in order to place the horizontal semicircular canal in a vertical position. The left external ear canal was irrigated for 2 min with 100 ml of cold water (20 °C). Typically, left cold CVS elicits a slow-phase nystagmus towards the left side and a fast-phase nystagmus towards the right side accompanied by an illusory feeling of self-motion to the right and predominantly activates the right hemisphere, where the left hand is represented (*Lopez, Blanke & Mast, 2012*). Importantly, the right hemisphere is the dominant hemisphere for vestibular function (*Dieterich et al., 2003*). Sham stimulation consisting of irrigating the left ear with water at body temperature (37°) was used as a control. The sham stimulation elicits similar extra-vestibular cues (e.g. tactile stimulation), but is widely viewed as not leading to an activation of the vestibular system (*Miller & Ngo, 2007*). Two experimenters verified the effectiveness of the stimulation by visually assessing the nystagmus. A vestibular stimulation questionnaire (adapted from *Lenggenhager, Lopez & Blanke, 2008*; *Lopez, Lenggenhager & Blanke, 2010*) assessed the subjective effects of vestibular stimulation. In particular, illusory motion was assessed on a numerical rating scale (NRS) from 0 to 6 (0 = no motion, 6 = very strong motion). A total of 13 other common symptoms associated with vestibular stimulation were also assessed on a NRS from 0 to 6 (0 = absent, 6 = severe). These were general discomfort, nausea, vertigo, racing heartbeat or palpitations, difficulty concentrating, drowsiness, faintness, sweating or cold sweat, need to vomit, headache, fatigue, pallor, and blurred vision (please see Figs. S2A and S2B). The vestibular stimulation questionnaire was administered before (in order to familiarise participants with the questionnaire and to clarify any clinical terms), and immediately after CVS and sham stimulation.

In line with other experimental protocols, adaptation and self-location testing with the MIRAGE system (see below) were conducted after the nystagmus had subsided, in order to prevent perceptual biases induced by the directional beating of the nystagmus itself. This is possible since CVS studies have revealed effects that outlast the duration of the stimulation as shown by several neuroimaging and behavioural studies (please see *Ferrè et al., 2015*, for a thorough discussion). Participants were able to rest for 30 min between the two conditions, to minimise a possible carry-over effect.

## Apparatus (MIRAGE multisensory illusion box) and experimental setup

The MIRAGE system consists of a two-chamber box with a double sided mirror between the bottom chamber and the top chamber, and a video screen facing down placed on

the ceiling of the top chamber. The system is arranged so that video images of the lower chamber are projected onto the mirror of the top chamber. Therefore, the mirror appears to show what is in the chamber below. The result is that, when the participants look down towards their hands, they see an image of their own hands at approximately the same location as the actual hands. Participants comfortably sit in front of the box while a bib blocks the view of their arms or any other visual cues about their hand position. Using customised software, the projected image is capable of being manipulated in order to create the visual illusion that the participants' hands were closer together than they actually were, thus making the visual and proprioceptive input incongruent (see Fig. 1). For more details on the MIRAGE, see https://miragelab.co.uk/.

## Experimental procedure

Participants first learned and practised a well-established localisation task (*Bellan et al., 2015*) with accurate visual information. They placed their hands into the MIRAGE box with a computer-generated red arrow projected at the midpoint between the hands while seeing an image of the hands at their actual location. The arrow moved towards the tested hand at a constant velocity (approximately 2.86 cm/s) and participants said 'stop' when the arrow was pointing at the image of their middle finger. Participants practised this localisation task at least six times while able to see an image of the tested hand in the MIRAGE box, or until participants were comfortable with their accuracy in performing the task. The tested hand was then hidden from view and participants repeated the localisation task by saying 'stop' when the arrow aligned with where they believed their middle finger to be; this was practised at least five times, or until participants were comfortable with the task. Throughout the experiment, the arrow always moved from the centre of the screen towards the unseen hand. The direction of movement of the arrow does not affect localisation judgments (*Bellan et al., 2015*). The time between practice trials was 15 s. After localisation practice for each hand, participants removed their hands from the box and completed a seven-item visual-analogue scale (VAS) questionnaire about their perceptions of the hand that was tested (see first column in Table 1). This provided an opportunity to clarify items on the questionnaire and familiarise participants with the process. Questions one and five were designed to assess self-location. This questionnaire was a shortened VAS version of that used in the original DHT experiment (*Newport & Gilpin, 2011*). We used VAS over NRS because of its superior psychometric properties (*Price, Staud & Robinson, 2012*).

During the actual experiment, the caloric or sham stimulation was performed as described above and, once nystagmus subsided, the participant walked to another, adjacent room where the localisation task was performed by a third experimenter, blinded to the condition (CVS or sham). Participants were instructed not to discuss or report symptoms to this experimenter and there was no conversation other than MIRAGE instructions and participant responses. First, the MIRAGE adaptation procedure was performed by having participants place their hands into the MIRAGE box, holding the hands approximately five cm above the floor of the box and not touching any portion of the box with their hands or forearms. During the adaptation procedure, participants were asked

**Table 1 Analysis of MIRAGE questions.**

| Questions | Friedman[§] | Post hoc Wilcoxon[†] |
|---|---|---|
| Self-location | | |
| 1. I knew exactly where my right[‡] hand was | $X^2 = 2.27, p = 0.519$ | |
| 5. I couldn't tell where my right[‡] hand was | $X^2 = 5.05, p = 0.168$ | |
| Sense of ownership | | |
| 2. My right[‡] hand was part of my body | $X^2 = 2.61, p = 0.456$ | |
| 7. It seemed that my right[‡] hand no longer belonged to me | $X^2 = 3.12, p = 0.372$ | |
| Self-location and sense of ownership | | |
| 4. It seemed like my right[‡] hand was no longer there | $X^2 = 8.52, p = 0.036*$ | L: $Z = -2.50, p = 0.012^{\#}$<br>R: $Z = -2.06, p = 0.039$ |
| Sensation of the hand | | |
| 3. The sensation in my right[‡] hand was more vivid than normal | $X^2 = 1.79, p = 0.688$ | |
| 6. I had the sensation that my right[‡] hand was numb | $X^2 = 7.23, p = 0.065$ | |

Notes:
[§] Friedman two-way ANOVA by rank.
[†] Wilcoxon signed rank test performed on the left (L) and right (R) hands separately.
[‡] Substituted the word 'left' for 'right' when the left hand was tested.
[*] Significant $p < 0.05$.
[#] Significant $p < 0.025$.

to keep their hands between two visible blue bars surrounding each hand. The visual image of the hands was manipulated so that the hands looked like they were gradually moving inwards. However, in order to keep the hands between the blue bars, participants were actually moving their hands further apart. At the end of the adaptation procedure, which lasted 25 s, the physical location of each hand was 150 pixels (approximately 13.65 cm) lateral to the visual image of the hand (see Fig. 1). The experimenter then placed the participants' hands on the floor of the box, where the hands remained until the first round of testing was complete.

Participants then performed the localisation task, by saying 'stop' when the arrow aligned with where they believed their middle finger to be. The localisation task was performed six times at 15-s intervals, and the stop position of the arrow recorded for each trial. After approximately every third trial, participants were verbally reminded to focus on where they felt their middle finger to be. The actual physical location of the middle finger was recorded by the experimenter. Participants then removed their hands from the MIRAGE box and completed the questionnaire about subjective experience of the hand that was just tested. This process was then repeated for the other hand. It is not known exactly how long the effects of CVS last, but prior research using CVS to modulate bodily sensations found changes that persisted for at least 15 min after the CVS (*Ferrè et al., 2013*). Consequently, the MIRAGE and localisation tasks were performed as quickly as possible after the caloric or sham stimulation and always within this 15-min window. Testing of both hands was completed in an average of 10 min 36 s (±54 s) after completion of the CVS or sham; the maximum time required for MIRAGE testing was 14 min from completion of the CVS or sham. Hand (right vs. left) and condition
(CVS vs. sham) were counterbalanced and randomised in blocks of four. After the first session, participants were able to rest for 30 min.

Finally, at the end of the experiment, participants were asked five debriefing questions to determine subjects' awareness of the illusion: (1) What do you think we were studying in the experiment? (2) Did you notice a difference between the two stimulation conditions? (3) What do you think was happening in the position tracking part (i.e. when the hands were between the two blue bars) of the experiment? (4) Were you aware that you were moving your hands during the position tracking part of the experiment? (5) If yes to previous question: In what direction were your hands moving?

## Statistical analysis

Hand location error was calculated as the difference between the participants' estimated position and the last seen position of the middle finger. Last seen position was used rather than the physical position because we were interested in initial localisation and speed of drift. To test this, a linear regression was performed on location at the six time points, resulting in an *intercept* (reflecting initial displacement relative to visual location) and *slope* (reflecting speed of drift from visual to physical or proprioceptive location) for each hand and condition. Slope and intercept were analysed using a repeated measures analysis of variance (ANOVA) with the factors Stimulation (CVS, sham) and Hand (left hand, right hand) as within-subjects factors. Responses to many of the MIRAGE questions were not normally distributed (and could not be transformed to be normal) and were therefore analysed non-parametrically using Friedman two-way ANOVA by ranks, with stimulation (CVS, sham) and Hand (left hand, right hand) as the four conditions. Post hoc analysis was done using Wilcoxon signed rank test for CVS versus sham stimulation; Bonferroni correction was performed for multiple post hoc tests ($p = 0.05/2$, as right and left hands were analysed separately). Wilcoxon signed rank test was also used to analyse the vestibular stimulation questionnaire (CVS versus sham). Statistical tests were performed with IBM SPSS Statistics 22.

To quantify how much the data should shift our belief in favour of the null or the alternative hypothesis, we computed Bayes Factors for the main analyses (BF10 where 1 means that the two hypotheses are equally likely, larger values indicate more evidence for the alternative hypothesis, and smaller values indicate more evidence for the null hypothesis; BF01 where values above 3 indicate substantial evidence for the null hypothesis). All Bayesian analyses were computed with JASP version 0.8.6 using Cauchy-distributed objective priors, centred on zero, with a default scale of 0.707 (i.e. 50% of possible standardised effect size values are expected to fall between −0.707 and +0.707) (*JASP Team, 2018*).

## RESULTS

### Hand localisation

As expected, localisation drifted from the last seen position outwards (i.e. where the hand actually was). Figure 2 illustrates the average location error for each of the four conditions separately. However, as indicated by the ANOVA for slope, this error did not

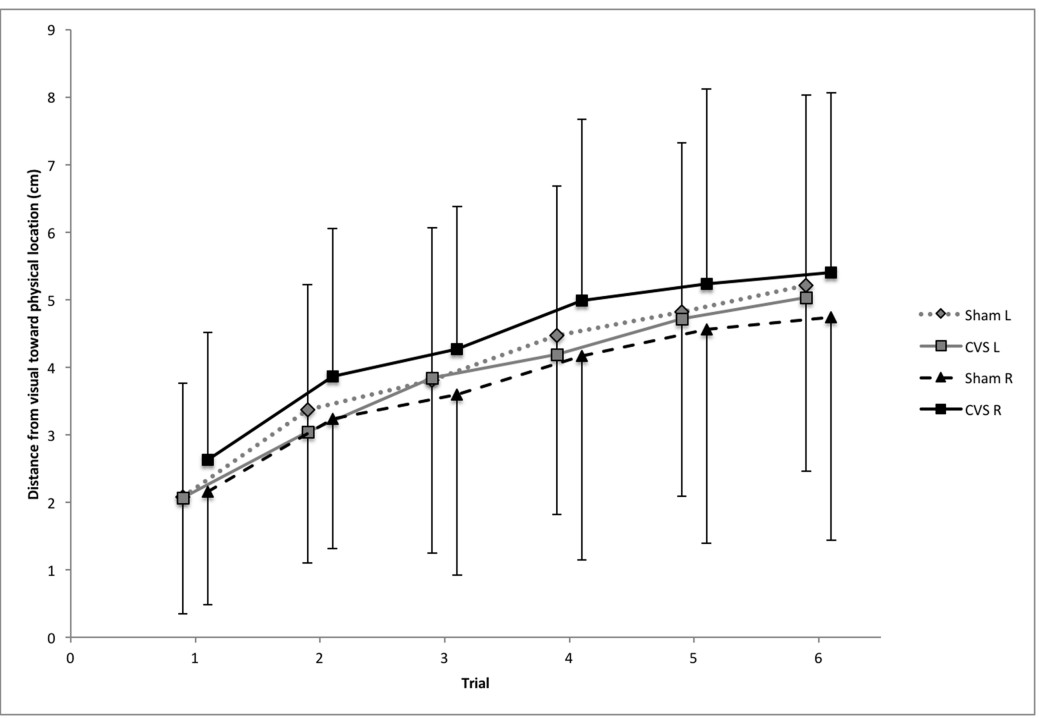

**Figure 2 Average drift.** Localisation of the middle finger, average for all participants at each of the six trials; standard deviation bars shown going up for CVS and down for sham. Slope and intercept were calculated based on these results. Data markers for left and right hands are shifted slightly left and right, respectively, of trial number to more clearly differentiate the data. Distance is measured from 0 = visual location to 13.65 cm = physical location of the middle finger.

significantly differ between the four conditions: there was no significant main effect of Stimulation ($F_{(1,23)} = 0.036$, $p = 0.852$, $\eta_p^2 = 0.002$, BF10 = 0.213, BF01 = 4.694), no significant main effect of Hand ($F_{(1,23)} = 1.016$, $p = 0.324$, $\eta_p^2 = 0.042$, BF10 = 0.399, BF01 = 2.504), and no significant interaction Stimulation × Hand ($F_{(1,23)} = 0.283$, $p = 0.600$, $\eta_p^2 = 0.012$, BF10 = 0.026, BF01 = 37.960).

The ANOVA for intercept also revealed no significant differences: there was no significant main effect of Stimulation ($F_{(1,23)} = 1.085$, $p = 0.308$, $\eta_p^2 = 0.045$, BF10 = 0.284, BF01 = 3.521), no significant main effect of Hand ($F_{(1,23)} = 1.601$, $p = 0.218$, $\eta_p^2 = 0.065$, BF10 = 0.643, BF01 = 1.556), and no significant interaction Stimulation × Hand ($F_{(1,23)} = 2.012$, $p = 0.170$, $\eta_p^2 = 0.080$, BF10 = 0.094, BF01 = 10.659).

Please note that the Bayes Factors support the null hypothesis.

## Questionnaire/Subjective reports of the MIRAGE

Hypothesis testing (Stimulation and Hand) showed that Question 4 ('It seemed like my right/left hand was no longer there') was significant ($p = 0.036$), with post hoc Wilcoxon tests for CVS vs. sham significant for the left hand ($p = 0.012$) but not the right hand ($p = 0.039$). None of the other questions revealed a significant difference between CVS and sham. See Table 1 for the statements of all questions and full statistical indices and please refer to Fig. S1 illustrating the results of the questionnaire.

## Vestibular stimulation questionnaire

### *Illusory motion*

To assess illusory motion due to CVS, we asked participants: 'Did you perceive any type of motion during the stimulation?' and, if they responded affirmative, they were questioned further about how strong this perception was for the whole body, upper body, head, arms, and, legs (please refer to the method section 'CVS' for more details on the questionnaire). All items were different between conditions (all $p < 0.01$), indicating that participants—as expected—experienced stronger illusory motion perception after CVS than after sham. Ratings were highest for head motion (mean 3.9 ± 2.2 SD) and lower for the arms (1.8 ± 2.0 SD) and legs (1.4 ± 1.9 SD). One participant did not report experiencing illusory motion, but nystagmus was observed and other symptoms reported, so we were confident that CVS was effective in this participant. All other participants reported the strength of illusory motion to be 2 or greater in the CVS condition.

We also asked participants to describe in what direction they perceived themselves to move (perception of illusory own body-motion). While many participants reported experiencing a circular motion to the right, a few participants found it difficult to clearly indicate the directional quality or found it difficult to distinguish between illusory own-body motion and illusory motion of the room (i.e. visual field).

### *Stimulation induced symptoms*

Despite some moderate general discomfort and other symptoms such as vertigo and dizziness, all participants were able to complete the experiment. One participant reported slight dizziness (between 1 and 2 on a scale from 0 to 6) up to five days later; while unusual, this long duration of effects was also observed once during a pilot test.

Participants frequently reported vestibular related side effects after CVS but seldom after sham (Figs. S2A and S2B). Most symptoms (10/13), namely, general discomfort, nausea, vertigo, racing heartbeat or palpitations, difficulty concentrating, drowsiness, faintness, sweating or cold sweat, need to vomit and blurred vision, were rated significantly higher (all $p < 0.05$) after CVS than after sham. Head tension or headache and fatigue were no more common after CVS than after sham ($p > 0.1$); pallor was not significant ($p = 0.07$). Most (>50%) participants reported general discomfort and difficulty concentrating, 50% reported vertigo and fewer reported other symptoms (please refer to Figs. S2A and S2B).

## Debriefing questions

Participants were not aware of the precise hypothesis tested but they all noticed a difference between CVS and sham, typically pointing out that they experienced more and stronger side effects during CVS than they did during sham (see section above). Nine participants hinted that they believed that we were testing how CVS *impairs* their ability to perform the task.

Most participants reported that their hands moved as needed to remain between the two blue bars, but either said that they were moving only slightly back and forth, or inwards

(i.e. towards each other). Four participants reported that their hands were in fact moving outwards (correct observation).

## DISCUSSION

We investigated whether CVS changes the relative weighting of proprioceptive and visual information in self-location, or bodily perceptions of the hands, when participants receive conflicting visual and proprioceptive information regarding hand position. Our results show that, although the feeling of the target hand (i.e. the hand that was out of view and had to be localised) is different after CVS, as compared to after sham, there was no effect on perceived location of the hand, nor on the weighting of proprioceptive and visual information during self-location. These findings are evidenced by no differential effect of condition on the slope or intercept of the location data, but a stronger reported feeling after CVS that the hand was 'no longer there' (main effect question 4 of the MIRAGE questionnaire). Post hoc testing showed that this main effect seems to be driven by the results of the left hand (significant result for the left hand and a trend only for the right hand) possibly reflecting the predominant activation of the right hemisphere after left cold CVS and/or the right hemisphere dominance of the vestibular system.

Our results appear in line with some previous literature and in contrast to other. In particular, vestibular stimulation did not affect the objectively measurable change in self-location in a RHI paradigm (i.e. the proprioceptive drift) (*Lopez et al., 2012*; *Lopez, Lenggenhager & Blanke, 2010*), but it did impact relevant subjective measures (i.e. increased illusory ownership of the rubber hand and the illusory location of touch) (*Lopez, Lenggenhager & Blanke, 2010*). Our results extend our understanding of vestibular effects by showing that bodily feelings that are closely aligned with proprioception (the feeling of the hand being there) could change independently of proprioception. Our findings are in contrast however with previous results reporting that GVS can reduce the drift (i.e. reduce mislocalisation of one's own hand) in a RHI paradigm, showing enhanced weighting of proprioceptive input through vestibular stimulation (*Ferrè, Berlot & Haggard, 2015*). Our data diverge also with studies in healthy participants that showed *impaired* proprioception after GVS (*Ponzo et al., 2018*; *Schmidt et al., 2013a*).

Participants were in general not aware of the mismatch between the seen and the physical position of the hand. For most participants, the MIRAGE illusion was effective in creating a mismatch between visual and proprioceptive localisation of the hands; only four participants were aware that their hands physically moved in a different direction from the visual image of their hands. This is important because it overcomes a limitation of the RHI, where participants cognitively know the rubber hand is not part of them and that their own hand rests out of view. Moreover, doubt has been raised over the validity of proprioceptive drift as a 'biomarker' of embodiment of the rubber hand. Proprioceptive drift occurs after the asynchronous hand stroking condition, also when the illusion is not induced, and even when no hand is displayed at all (*Rohde, Di Luca & Ernst, 2011*). Data from modified versions of the MIRAGE multisensory illusion indirectly support the idea that proprioceptive drift observed during the RHI is more likely to reflect visual capture of touch than embodiment of the artificial limb (*Bellan et al., 2017a*, *2015*).

Participants in this study did not report perceptual changes that have been observed during the RHI after CVS, but did report perceptual changes reflecting absence or numbness of the hand. This raises the very important possibility that such perceptual reports reflect expectation bias. It is intuitively sensible that, when an illusion replaces ones' own hand with a rubber counterpart, the rubber hand might feel 'owned'. In contrast, when the current study removed the tested hand from view during localisation testing, participants reported that the hand 'was no longer there' instead of lacking of ownership. This may simply reflect the importance of visual confirmation on body localisation.

A few participants in this study reported that they believed that we were testing how CVS *impairs* their ability to perform the task and no one expected it might improve their ability. It therefore seems crucial that participants were not aware of the task characteristics (mismatch in hand localisation). However, individual differences were considerable (as indicated by the large variance of localisation estimates) and future studies may benefit from identifying sources of inter-individual variability. Moreover, our results suggest that right-handers respond similarly to the mismatch in hand-location independent of whether their dominant or non-dominant hand disappeared. In fact, to our knowledge, this is the first report using the MIRAGE to assess self-location bilaterally; we found right and left (dominant and non-dominant) hands had similar initial location relative to last observed visual location and similar drift from visual to proprioceptive (i.e. physical) location. Also, as predicted, we did not observe potential vestibular induced attentional effects as reflected by non-significant interactions between the two factors Stimulation and Hand. The corresponding Bayes Factors support the acceptance of the null hypothesis.

Differential effects on implicit (self-location) and explicit (what does the body part feel like) markers would not be predicted on the basis of the proposal (*Moseley, Gallace & Spence, 2012*) and clinical framework of the cortical body matrix (CBM) (*Bellan et al., 2017b*; *Wallwork et al., 2016*). The CBM proposes that a multimodal neural network subserves the regulation and protection of the body and the space around it, at both a physiological and perceptual level. A broad range of findings support the CBM idea: for example, limb-specific disruption of thermoregulation (*Moseley et al., 2008*) and altered histamine reactivity (*Barnsley et al., 2011*), induced by the RHI; efficacy of the RHI modulated by hand temperature (*Kammers, Rose & Haggard, 2011*); thermoregulation and RHI efficacy disrupted by TMS over multimodal posterior parietal cortex (*Gallace et al., 2014*); spatially-defined disruption of thermoregulation, ownership, pain (*Moseley, Gallace & Iannetti, 2012*), bodily perception (*Reid et al., 2016*) and motor performance (*Reid et al., 2018*). The compelling nature of that research adds weight to our earlier suggestion that the pattern of changes in how the body feels without changes in how the body is regulated potentially reflect reporting bias. Of course, this issue needs to be interrogated more fully, but we contend that null results in rigorous studies powered a priori, such as this one, are critical for this process. The Bayes Factors support the null hypothesis. These indices strengthen the idea that the null hypothesis may indeed be more likely than the alternative hypothesis, arguing against a falsely negative finding.

Although CVS does not improve self-location of a body part, it remains possible that it would improve full body location. While self-location is the term typically used in such

contexts, it may be more accurate to describe 'body part location with respect to the self' (*Blanke & Metzinger, 2009*; *Lenggenhager & Lopez, 2016*). When using an embodied mental transformation task (often also referred to as egocentric mental rotation task), CVS appears to improve performance for whole bodies, but not for single body parts (i.e. hands) (*Falconer & Mast, 2012*). Even though people with bilateral vestibular loss are worse than healthy participants at embodied mental transformations, the impairment is not affected by whether the whole body or body-parts are involved (*Grabherr et al., 2011*; see also *Wallwork, Butler & Moseley, 2013*). Clearly more research is required. While technically more demanding to investigate full-body illusions compared to body-part illusions, the use of visuo-vestibular cue manipulation to induce and manipulate full-body illusions (*Macauda et al., 2015*; *Preuss & Ehrsson, 2018*) appears a step in the right direction.

As pointed out in the introduction, research in patient populations suggests that vestibular stimulation can enhance bodily self-consciousness. For example, phantom limb pain decreases after CVS (*André et al., 2001*). The authors interpreted that finding by suggesting that CVS enhances both body awareness and phantom limb sensation (by replacing a deformed with a normal representation). The current study was not able to similarly enhance bodily self-consciousness among healthy individuals. This may reflect a fundamental difference in vestibular effects between people with clinically impaired bodily self-consciousness and healthy controls. In fact, vestibular stimulation has been reported to enhance proprioceptive localisation (arm position sense) in patients with impaired bodily awareness (i.e. spatial neglect), but impair it in healthy controls (*Schmidt et al., 2013b*). A recent study investigating potential beneficial effects of CVS in patients with a body identity integrity disorder, however, reported a null result (*Lenggenhager et al., 2014*), suggesting that not all conditions are benefitted by CVS. The background of possible publication bias and widespread promotion of vestibular stimulation benefits also points to the importance of publishing null results such as those found here (for a discussion see *Grabherr, Macauda & Lenggenhager, 2015*; for other recent null results in healthy participants see *Macrea et al., 2016* and patients see *Deroualle et al., 2017*).

One potential limitation of the study is inadequate blinding of the experimenter collecting MIRAGE data, even though we explicitly avoided any conversation regarding symptoms. While this blinding may not have been sufficient in some cases due to possible observable effects induced by CVS (e.g. pallor), error introduced by inadequate blinding would most likely have increased differences in MIRAGE results as that was the hypothesis being tested. It is unlikely that inadequate blinding would have incorrectly led to the negative result observed. Future research should attempt to assess and further improve blinding, if possible.

Another potential limitation is the possible influence of head movements due to walking in-between stimulation and MIRAGE testing. Due to the required head tilt to deliver CVS, many studies have their participants' head and/or body position readjusted to an upright sitting position after CVS (*Ferrè et al., 2013*; *Karnath, Himmelbach & Perenin, 2003*; *Ngo et al., 2007, 2008*) allowing to conduct the subsequent experimental condition in a more natural and to other studies comparable head and body position. Please note that

typically during these experimental conditions, the participant's head is not restrained allowing head motion (see for example *Karnath, Himmelbach & Perenin, 2003*). Moreover, studies involving patients revealed long lasting CVS induced effects that must have 'survived' the effects of walking (*André et al., 2001*; *Aranda-Moreno et al., 2019*). Taken together, the results of these studies show that there are CVS induced effects that outlast primary effects such as nystagmus and that they are at least to a certain extent robust to head movements. However, a subtler or more specific impact of movements cannot be fully ruled out due to the vestibular system's inherent multisensory nature and we are not aware of a study precisely addressing this question (contrary to several studies addressing the question how vestibular stimulation affects walking). Yet, as outlined above, there is evidence that walking would not completely abolish these longer lasting CVS effects. For our protocol, we estimated that the advantage—the blinding of the experimenter without losing time (the MIRAGE experimenter was set-up and ready to go as soon as the participant came in)—outweighed this potential disadvantage.

Galvanic vestibular stimulation may be better suited than CVS to assess vestibular contributions to self-location because it can be applied in a more precise fashion (*Palla & Lenggenhager, 2014*) although compelling progress has been made improving CVS devices (*Black et al., 2016*). Although we undertook pilot studies and based our protocol on the available literature in the field, it remains possible that our CVS procedure artificially induced a vestibular input that was too strong to observe a beneficial effect (i.e. the aversiveness of the stimulation may cancel out the beneficial effects). Vestibular stimulation is often induced in experimental studies, as in this one, in a dichotomous fashion: it is either provided (experimental condition) or not (control condition). We have argued elsewhere that future experiments may need to consider more fine-tuned parameters taking into account the complexity and highly nuanced functioning of the vestibular system (*Grabherr, Macauda & Lenggenhager, 2015*). We used double-verification of nystagmus (visual observation by a physician and a vestibular specialist), but did not include an objective, quantifiable nystagmus measure. This approach may have left us unable to identify participants with late-onset of nystagmus or a weaker response, both of which could possibly affect our results.

Our final consideration is that we did not lodge and lock our experimental protocol prior to conducting this experiment. Doing so enhances the transparency of research and is now recommended practice for observational and clinical designs in many fields and including this step would have enhanced the confidence with which the reader could accept our results (*Lee et al., 2018*).

## CONCLUSIONS

This investigation of whether CVS changes the relative weighting of proprioceptive and visual information in self-location, or bodily perceptions of the hands, showed that although the subjective feeling of the hands appeared to be disrupted by CVS, there is no effect on self-location of the hand, nor on the weighting of proprioceptive and visual information during self-location. The methodological advantages of the MIRAGE

procedure, and the naivety of participants to the illusory manipulation, add weight to the finding that CVS does not improve self-location of a single body part.

## ACKNOWLEDGEMENTS

We would like to thank Bigna Lenggenhager for her valuable advice on the manuscript and Cat Jones for the pictures used in Fig. 1.

### Funding

Luzia Grabherr was supported by the Swiss National Science Foundation PBBEP1-144848; G. Lorimer Moseley was supported by the National Health & Medical Research Council of Australia ID 1061279; this work was supported by a NHMRC Project Grant awarded to G. Lorimer Moseley ID 1008017. The funders had no role in study design, data collection and analysis, decision to publish, or preparation of the manuscript.

### Grant Disclosures

The following grant information was disclosed by the authors:
Swiss National Science Foundation: PBBEP1-144848.
National Health & Medical Research Council of Australia: 1061279.
NHMRC Project: 1008017.

### Competing Interests

In the last 5 years, G. Lorimer Moseley has received support from: Pfizer Australia, Kaiser Permanente, Workers' Compensation Boards in Australia, Europe and North America, AIA Australia, the International Olympic Committee, Port Adelaide Football Club and Arsenal Football Club. Professional and scientific bodies have reimbursed him for travel costs related to presentation of research on pain at scientific conferences/symposia. He has received speaker fees for lectures on pain and rehabilitation. He receives book royalties from NOIgroup publications.

### Author Contributions

- Luzia Grabherr conceived and designed the experiments, performed the experiments, analyzed the data, prepared figures and/or tables, authored or reviewed drafts of the paper, approved the final draft.
- Leslie N. Russek conceived and designed the experiments, performed the experiments, analyzed the data, prepared figures and/or tables, authored or reviewed drafts of the paper, approved the final draft.
- Valeria Bellan authored or reviewed drafts of the paper, approved the final draft.
- Mohammad Shohag performed the experiments, approved the final draft, performed the otoscopy.
- Danny Camfferman conceived and designed the experiments, performed the experiments, approved the final draft.

- G. Lorimer Moseley conceived and designed the experiments, contributed reagents/materials/analysis tools, authored or reviewed drafts of the paper, approved the final draft.

## Human Ethics

The following information was supplied relating to ethical approvals (i.e. approving body and any reference numbers):

The Human Research Ethics Committee of the University of South Australia granted Ethical approval to carry out the study within its facilities (Application ID: 32955).

## Data Availability

Raw data is available in File S1.

## Supplemental Information

Supplemental information for this article can be found online at http://dx.doi.org/10.7717/peerj.7201#supplemental-information.

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
