# Peer review of "The disappearing hand: vestibular stimulation does not improve hand localisation"

_PeerJ, doi:10.7717/peerj.7201_

## Round 0.1 · original submission · Minor Revisions

I have now received three reviews on your paper, and have carefully read the manuscript myself. FIrst of all, I want to apologize for my delay in getting back to you. I also want to thank the reviewers for their work. As you will see, the reviews are very constructive, clear and helpful, so please follow them for details. In general, I agree with the reviewers according to which the paper is in good shape, almost ready for publication. You will need to revise a bit the introduction, explaining the description of the illusion and rendering it sharper in some points (see comments by Rev 1 and 2), to address the issue of whether walking between the stimulation and the task execution might influence the results (see comment of Rev 3), to refine the analyses both providing an interpretation the Bayesian analyses and applying corrections to multiple comparisons (see comments of Rev 2 and Rev 3).

·

Basic reporting

I find the manuscript clearly written. Sufficient background and reference to previous literature is provided. I have especially appreciated the fact that three competing scenarios are presented to the reader: each is properly detailed, supported by references, and maps unequivocally to specific outcomes of the main task. The structure of the paper is adequate, and does not depart from what is customary in the field. The raw data is provided in the form of an excel file, which I could assess, with appropriately labeled headers; replication of the analyses is thus in principle possible. All the relevant results from the experiment, as far as I can tell, have been described in the main text/supplementary materials.

1) I only have a minor suggestion concerning the introduction: lines 71-76 appear rather vague. For the benefit of readers unfamiliar with the DHT, the authors may consider rephrasing this passage along the lines used in the methods (e.g. 177-180). Moving Figure 1 about here may also help.

2) Figures are appropriate (however, Figure 2 uses cm as units, whereas data in Table 1 is given in mm).

Experimental design

The research falls within the scope of the journal. The research question is important and well identified. The methods are sound. Statements concerning ethics approval are given.

Only a few details in the methods are missing, and in my opinion should be reported:

3) Line 127: power analysis. Justification for choosing an a-priori effect size of 0.3 should be given, for example by referring to previous similar research.

4) I commend the authors for using a complementary Bayesian approach. This approach is particularly interesting in order to provide support for H0 (instead of a mere failed rejection). However, please consider the following points:

a) Bayes factors are appropriately introduced in lines 258-262 and reported for the main tests (270-278). However, they are not interpreted. That Bayes Factors generally support H0 should be, instead, mentioned in the results section, and perhaps rehearsed in the discussion (e.g. around line 408, when discussing the importance of a-priori well powered studies capable of supporting H0). Ultimately, these indices strengthen the idea that H0 may indeed be more likely than H1, arguing against a falsely negative finding.

b) Because all the BFs favor H0, the authors may consider switching to Base 1 BFs (obtained as the reciprocal to 1 of BF10). This would make more intuitive assessing the degree of support for H0. For example, a BF10 of 0.123 would become 1/0.123= 8.13, indicating that the probability of obtaining the data would be about 8 times more likely under H0 with respect to H1.

c) Bayesian analyses are, to some extent, prior-dependent. Thus, the prior distribution used should be described. I believe JASP (which, by the way, should be referenced: JASP Team (2018). JASP (Version 0.8.6)[Computer software].) uses “default” Cauchy-distributed objective priors, centered on 0, and with a scale parameter of 0.707 (i.e. 50% of possible standardized effect size values are expected to fall between -0.707 and + 0.707).

Validity of the findings

The conclusions capitalize on negative results; however, such null findings can be interpreted considered the a-priori power analysis and the Bayesian statistical approach. Raw data is provided for anyone to replicate/expand these findings. Several possible limits of the study have been identified and discussed. Conclusions are consequent to the results, well stated, and when speculative appropriately identified as such.

5) The control (sham) condition is appropriate though far less unpleasant than the active CVS condition; this can be hardly avoided, as distress is to some extent inherent to a vestibular perturbation. However, one may wonder how effective the blinding procedures (line 205) would be under such strikingly different after-effects. Can the authors elaborate about this?

6) I must (partially) disagree on the authors’ assessment in lines 454-457 – “we did not lodge and lock our experimental protocol (…) failure to do so is a shortcoming of this work”.
I am myself a fierce defender of open practices, including pre-registration. Pre-register and possibly peer-review the protocol before data collection (i.e. registered report article format) is, I find, an effective way to improve methodological practices in one field, and counter-act publication bias by favoring methodologically sound studies regardless of their outcome. This is also possible and it has been done recently with vestibular stimulation protocols.
However, while such formats may inspire further confidence in the results, I don’t think that any study that is not formally pre-registered is necessarily inferior, or has shortcomings. This would be a radical opinion and I don’t think it has been ever stated this way. Instead, shortcomings arise in case of undisclosed flexibility in data acquisition or statistical approaches. For example, false positives are inflated for data peeking that leads to multiple sequential analyses and thus affects the sampling plan (i.e. stop data collection as soon as one test is significant, or continue data collection if one test is *almost* significant). Other degrees of freedom of the researcher – removing outliers, change statistical approach, etc. – also yield increased risk of false positives. All research is characterized by the presence of several arbitrary steps, and this is not eliminated by pre-registration; what pre-registration tackles is the transparency in which a researcher is forced to report all these choices, which are in this context undertaken independently of the results.
In this study: a sampling plan was established a-priori following power analysis; one participant was, indeed, discarded, but this has been clearly disclosed and motivated (and her/his data reported in the supplementary materials); analyses seem to be rather customary for the task. Ultimately, I suggest the authors to re-consider this point.

Additional comments

This study is important and timely. I am positive it will provide food for thoughts for many researchers in the field.

Reviewer 2 ·

Basic reporting

This experimental work exploits a perceptual illusion that gives participants conflicting information about their hands position (proprioceptively- vs. visually-encoded). The authors’ hypothesis is that caloric vestibular stimulation would improve hand localization enhancing the proprioceptive location over the visual one. The underlying aim of the study is to expand the understanding of the role of vestibular information in bodily self-consciousness. The literature review informs the basis of the research itself, since contrasting findings exist regarding the effects of vestibular stimulation on self-location mechanisms and body-ownership feelings.

Overall, I think that this work makes an interesting contribution to the field. For the improvement of the paper I have the following suggestions:

The introduction section needs to be improved including a straightforward presentation of your research question and experimental work.

Lines 71-80: While the rubber hand illusion procedure is well known, the disappearing hand trick is far less known. I think it would help the reader to spend a few more words on how the mirage box practically works. Also, being the drift one of your measurements, its description needs to be expanded.

Lines 91-94: in this section it seems that you’re making predictions about the possible effects of CVS without having explicitly introduced your experiment and measurements. Also, mentioning slope and intercept here (as well as in lines 104-106) is confusing.

Lines 107-111: This paragraph is vague. If you believe that CSV induce spatial attention effects, then you should clarify in which direction these effects are expected to affect your results, and discuss this prediction in the results and discussion section.

The information that you’re testing both hands is missing. Do you have any specific prediction about the effects of CVS on the initial localization of left and right hand/drift?

Line 68: missing reference for the evidence that proprioceptive drift is correlated with the strength of the illusion (e.g., Tsakiris and Haggard 2005).
Line 75; 80; 373: Bellan et al. 2017 a or b?

In lines 82 and 299 you mention sections that seem not to be present in the manuscript.

Please, make the symbols in Table 2 uniform.

The whole conclusions section repeats lines 458-464!

Experimental design

Material and methods are well described, but in the experimental procedure section some details should be restructured. For example, initially it is not clear whether you’re testing only one or both hands. Also, in lines 193-194 the sentence “The direction of movements of the arrow does not affect localization judgments” doesn’t help with clarity and appears to be out of place. This statement needs to be moved in a different position and expanded, for clarity, or deleted.

Validity of the findings

The conclusions are appropriate given the results.

Reviewer 3 ·

Basic reporting

The article is written in professional English and in a clear and unambiguous manner. Literature review is extensive and includes the most relevant studies on the topic. The article is self contained with results addressing the stated hypothesis. Raw data are shared and the file is clear and easily readable.
Table 1 could be improved by explaining in the table caption why "mm/trial" is indicated only for the slope (the reader not very familiar with the task would have otherwise to go back to statistical analysis of the manuscript and infer why).
Table 1 uses mm but Figure 2 uses cm. Also, I am not sure both figure 2 and table 1 are needed: they are describing very similar parameters.
A similar reasoning on possible duplication applies to Figure 3 and Table 2. I wonder if the number of tables and figures could be reduced without impacting the quality of the manuscript.

Experimental design

The research question is well defined, and the authors did a particularly good job in stating why the research question is important (i.e. informing clinical studies).
The methods are well described and definitely allows replication of the study. Ethic standards are respected.
The chosen method is a valid and well recognized one.

There is 1 concerns about the methods reported in this study, for the CVS in particular.
The authors state that "during the actual experiment, the caloric or sham stimulation was performed as described above and, once nystagmus subsided, the participant walked to another, adjacent room where the localisation task was performed by a third experimenter, blinded to the condition (CVS or sham).". Walking after a vestibular stimulation actually disturbs the effect of the CVS itself. The different head position and the stimulation due to head movements of the vestibular receptors likely restored the system functioning. I am wondering if the authors can back up their choice and/or can discuss the impact of this experimental choice.

Validity of the findings

Data analysis is sound. However, for the post hoc comparisons, authors did not apply any multiple comparison correction. This could lead to an increase of false positives. It is worth considering applying a Bonferroni correction to the comparisons reported and see if they survive the corrected threshold. This would impact how the authors interpret their results on the subjective report and laterality.

In general, the results of this study are of interest to better understand the contribution of the vestibular system to the development of a sense of the body.
However, given the issue with the experimental design explained above, if walking disrupted the effect of CVS, then I am afraid the reported results are simply due to the fact that individuals were not under the effect of the stimulation anymore when carrying out the task.
Until this is ruled out, I am concerned about considering the conclusions valid or not.

Additional comments

The paper is well written, clear and with an attention to details that clearly indicated an effort towards allowing replication and having a reliable results, which overall is commendable given all the discussion on good practices in psychology which e are witnessing nowadays.
I think the authors need to carefully consider the effects of walking between the stimulation and the execution of the experimental task, and secondly they should consider protecting themselves from false positives in multiple comparisons.
Once these 2 points are cleared, i think the paper would provide a much stronger contribution to the literature.

---

## Round 0.2 · accepted · Accept

I am happy to inform you that your manuscript has been accepted for publication on PeerJ.

# ·

Basic reporting

-

Experimental design

-

Validity of the findings

-

Additional comments

The authors addressed all my points in a satisfactory way.

Reviewer 3 ·

Basic reporting

no comment

Experimental design

The point on the effects of walking on CVS is now addressed and clearly the experimental choice is backed up. The addition in the manuscript will be helpful should a reader have the same doubt.

Validity of the findings

Thanks for taking into account the potential issue of false positives. I think that the use of corrections for multiple comparisons make your findings stronger. Indeed the result and the corresponding explanation make sense.

Additional comments

I would like to thank the Authors for carefully considering my comments and addressing the concerns, both in their letter and in their manuscript.